# Identification and Characterization of ERK2 Dimerization Inhibitors by Integrated In Silico and In Vitro Screening

**DOI:** 10.3390/ijms262311481

**Published:** 2025-11-27

**Authors:** Carmen Ortiz-González, Berta Casar, Rafael Gozalbes, Eva Serrano-Candelas, Piero Crespo, Laureano E. Carpio

**Affiliations:** 1MolDrug AI Systems SL, Parque Tecnológico de Valencia, 46980 Valencia, Spain; cortiz@moldrug.com (C.O.-G.); rgozalbes@moldrug.com (R.G.); 2Instituto de Biomedicina y Biotecnología de Cantabria (IBBTEC), Consejo Superior de Investigaciones Científicas (CSIC), Universidad de Cantabria, 39011 Santander, Spain; berta.casar@unican.es; 3ProtoQSAR SL, Parque Tecnológico de Valencia, 46980 Valencia, Spain; eserrano@protoqsar.com; 4Centro de Investigación Biomédica en Red de Cáncer (CIBERONC), Instituto de Salud Carlos III, 28029 Madrid, Spain

**Keywords:** ERK1/2, molecular modelling, ERK dimerization, MAPK/ERK inhibitors

## Abstract

Protein–protein interactions (PPIs) take place in many cellular processes, including the activation of cellular cascades, such as the MAPK/ERK (Mitogen-Activated Protein Kinase/Extracellular-Regulated Kinase) pathway. Deregulation of these pathways leads to the development of diseases, such as cancer. DEL-22379 is an ERK2 dimerization inhibitor, which presents anti-tumoral effects, without affecting ERK2 phosphorylation. Our aim was to identify new therapeutic molecules targeting ERK2 dimerization, based on DEL-22379 structure. In this study, we implemented a combination of computational and experimental workflow, which includes in silico techniques, such as scaffold hopping and virtual screening to generate a dataset of candidate compounds, a native PAGE (PolyAcrylamide Gel Electrophoresis) electrophoresis to experimentally screen the potential inhibitors, and a detailed molecular docking and chemical profile prediction to understand the potential mechanism of action of the selected compounds. From an initial dataset of 536 compounds, we obtained two hit molecules that exhibited inhibitory effects on ERK2 dimerization: Drug73 and Drug120. A computational analysis of the mechanism of action, unveiled that Drug73 and Drug120 presented an improved docking score, and better drug-like properties when compared to DEL-22379. This study shows that computational studies, in combination with experimental evaluation, can be useful and efficient to find new therapeutic compounds.

## 1. Introduction

Protein–protein interactions (PPIs) are fundamental regulators of cellular processes, including cell proliferation, survival, differentiation, and migration [1,2]. In pathological contexts such as cancer and chronic inflammation, aberrant PPIs rewire signaling networks and sustain hallmarks of disease, promoting uncontrolled cell growth, invasion, and resistance to apoptosis [3]. The mitogen-activated protein kinase (MAPK) cascade exemplifies the central role of PPIs in pathological signaling. Within these pathways, the cascade mediated by extracellular signal-regulated kinases ERK1 and ERK2 integrates signals from upstream activators to regulate transcriptional programs and cytoplasmic substrates, and their dysregulated interactions contribute directly to oncogenic transformation and metastatic dissemination [4,5,6].

Traditional pharmacological approaches to kinase inhibition have largely focused on targeting catalytic domains, most commonly the ATP (adenosine triphosphate)-binding site. However, this strategy frequently results in limited selectivity and the onset of resistance [7]. In contrast, interfering with PPIs has emerged as an attractive therapeutic strategy, since it allows modulation of non-catalytic functions that are often isoform- or context-specific [8]. By disrupting protein assemblies rather than enzymatic activity, PPI inhibitors may provide a more selective means of intervention with reduced off-target toxicity and a lower risk of resistance [9]. This paradigm has opened new avenues for targeting the MAPK/ERK pathway in cancer and inflammation, emphasizing the importance of systematically identifying modulators of ERK2 interactions.

Following activation by MAPK/ERK kinase (MEK)-mediated phosphorylation, ERK2 translocates to the nucleus, where it phosphorylates nuclear targets. At the same time, MEK-mediated phosphorylation promotes ERK2 dimerization, an essential prerequisite for ERK autophosphorylation [10] (Figure 1).

ERK2 dimers have been described to play an essential role in the activation of cytoplasmic substrates, such as Ribosomal S6 Kinase 1 (RSK1), and are essential to modulate cell motility programs [11]. Dimerization is both necessary and sufficient to drive cellular migration in mammary tumor cells, as shown by loss-of-function (chemical and genetic) and gain-of-function (constitutively dimeric ERK2) experiments [11]. Consistently, pharmacological or genetic blockade of ERK2 dimerization curtails tumor progression and metastatic traits without affecting ERK2 phosphorylation, whereas enforcing ERK2 dimerization fosters invasive phenotypes [11,12,13].

Control by scaffold proteins is central to this mechanism: kinase suppressor of Ras 1 (KSR1) facilitates ERK dimer formation and tunes its pro-migratory outputs; consistently, high KSR1 expression correlates with adverse metastatic features in clinical samples [11]. Independent evidence links KSR1/ERK signaling to epithelial–mesenchymal transition (EMT)-like reprogramming and invasive behavior, reinforcing the scaffold-dependent control of motility. Together with earlier work highlighting ERK dimer/scaffold partnerships [14] and isoform biology in the MAPK module [15], these findings position ERK2 dimerization as a proximal regulator of cytoskeletal remodeling, migration, and tumor dissemination.

Cell motility is not restricted to tumor invasion but also underlies essential processes such as immune cell migration, fibroblast migration, and wound healing. Aberrant activation of ERK signaling has been linked to pathological inflammation and tissue remodeling, where excessive motility of fibroblasts and immune cells contributes to disease progression [12].

Selective inhibition of ERK2 dimerization, while preserving upstream phosphorylation, represents a promising strategy to limit pathological migration without abolishing essential ERK functions. This concept has been supported by studies showing that disruption of dimerization reduces cytoplasmic ERK signaling and maladaptive cellular responses, while maintaining nuclear survival pathways [10].

The first small molecule described to selectively inhibit ERK dimerization was DEL-22379, which blocks ERK dimer formation without interfering with its phosphorylation. As a result, nuclear targets remain unaffected, while cytosolic functions of ERK are impaired [12].

DEL-22379 belongs to the 3-arylidene-2-oxindole chemical class and has demonstrated significant anti-tumor efficacy in RAS/ERK-driven cancer models [12]. Importantly, its effects are context-dependent, showing differential outcomes in tumors depending on its RAS-ERK pathway mutational signature, underscoring the need to understand signaling background when targeting dimerization [12].

Nevertheless, DEL-22379 has been demonstrated to be cell-unspecific, showing cytotoxic effects in tumor cell lines but also in primary cells [16], and some toxicity in vivo, specifically in liver and intestinal epithelia [12].

Overall, DEL-22379 provides a real case demonstrating that pharmacological disruption of ERK dimerization is feasible, effective, and selective, establishing precedent for the development of next-generation inhibitors. Yet, the field still lacks a broader set of small molecules acting by this mechanism, and the context-dependent effects of DEL-22379 in different oncogenic backgrounds underscore the need for new candidates and, importantly, for a deeper mechanistic understanding of how small molecules modulate ERK2 dimerization.

In this work, we combined scaffold-hopping with a focused virtual screening approach to guide the identification of new ERK2 dimerization inhibitors. Experimental testing allowed us to prioritize several promising candidates, which were subsequently analyzed in detail through refined docking at the monomeric ERK2 interface, representing the biologically relevant pre-dimerization state. This integrated experimental–computational workflow provided insights into how small molecules can interfere with dimer formation through distinct binding topographies and helped to elucidate the binding mechanisms underlying dimerization inhibition. Going beyond the mere identification of promising hits, our study aimed at establishing a functional framework to explore their potential as therapeutic modulators with relevance in both cancer and inflammatory contexts.

## 2. Results

### 2.1. Scaffold Hopping

The scaffold hopping technique is used to generate new chemical compounds from a known effective molecule by changing its chemical scaffolds but maintaining its function [17,18]. This way, hit compounds can be optimized by replacing scaffolds that can improve properties such as ADMET (Absorption, Distribution, Metabolism, Excretion, and Toxicity) and bioactivity, but maintaining essential functional groups for interaction with the target. In this case, DEL-22379 already presented inhibitory effects of ERK2 dimerization; thus, we applied this technique in order to optimize its structure. As a result, we obtained 536 derivative compounds, in which we can differentiate compounds that substitute the indoline-2-one group, others that change the piperidine group, and some that replace the indole core (Figure 2).

### 2.2. Virtual Screening

By these means, we obtained a group of DEL-22379-derived compounds on which we performed a virtual screening by molecular docking approaches to evaluate which of these showed affinity for the ERK2 structure, evaluating five poses for each ligand.

As ERK2 H176 is a key residue for DEL-22379 binding to the ERK2 molecule [12], we first filtered out all the compounds that could not possibly exert this interaction. Thus, we measured the distance between each ligand and H176, restricting our selection to the compounds that were located within a maximum distance of 5 Å from this residue. This filter resulted in reducing the number of potential hits to 300 compounds, which were subjected to further experimental evaluation in order to identify which of these were able to inhibit ERK2 dimerization.

The ligand with the best affinity for ERK2 presented a docking score of 13.084 kcal/mol (Figure 3), while most of the values ranged between 8 and 12 kcal/mol. Therefore, the binding affinity alone could not allow us to discriminate between active and inactive compounds. The similarity of these compounds with respect to DEL-22379 was measured using the Tanimoto index. The similarity scores of most of these new structures are above 0.5, which denotes a high similarity with respect to the original structure (Appendix A). This high similarity is potentially produced due to the preservation of the main structural cores of DEL-22379, while only modifying some regions of the molecule.

### 2.3. Experimental Evaluation

Among the identified putative ERK2 binders from the in silico analysis, the commercially available compounds (Appendix A) were subjected to experimental evaluation in order to identify candidate compounds. A native PAGE was performed to evaluate whether the tested ligands could inhibit ERK2 dimerization (Figure 4). In epidermal growth factor (EGF)-treated cells, ERK2 undergoes dimerization, as shown by the high electrophoretic mobility band. Dimerization is prevented by the addition of DEL-22379, which serves as a positive control.

In cells treated with DEL-22379, only the lower mobility band was evident, indicating that only ERK2 monomers are present (monomer/dimer ratio of 5.36). The same happens in the case of cells treated with Drug21, Drug73, and Drug 120, with a monomer/dimer ratio of 5.42, 4.6, and 4.25, respectively. However, in the case of Drug21, phosphorylation was impaired, as can be seen with the pERK/ERK ratio of 0.25, which is half of the value of the DEL-22379 ratio, indicating that its inhibitory effect on ERK dimerization is a potential consequence of preventing ERK2 phosphorylation. Contrarily, Drug73 and Drug120 preserve ERK2 phosphorylation, with ratios of 0.7 and 0.8, respectively, which were higher than DEL-22379, demonstrating that their inhibitory effect on dimerization is exerted by other means.

Interestingly, contrary to DEL-22379, Drug81 and Drug88 behave as promoters of ERK2 dimerization, as demonstrated by the prominence of the band corresponding to ERK2 dimers in the cells treated with these compounds. In these cases, ERK phosphorylation is maintained, even enhanced (Figure 4).

### 2.4. Detailed Docking of Candidates

The experimental results of the previous section unveiled two candidate compounds, Drug73 and Drug120, that could act as effective ERK2 dimerization inhibitors without affecting its phosphorylation. Thus, to better understand the potential mechanism of action of these ligands on ERK2, in-depth in silico studies were carried out.

As the DEL-22379 binding site seems to be located in the dimerization interface, we performed a flexible molecular docking calculation directed to this region’s residues (specifically, H176, F181, F329, E332, L333, D335, L336, P337, E339, K340, and E343). We clustered the obtained poses for each ligand, obtaining 13, 15, and 11 different clusters for DEL-22379, Drug73, and Drug120, respectively. Although the three compounds act as ERK2 dimerization inhibitors, their mean binding energies present differences (Figure 5). DEL-22379 binding affinities range between 7.5 and 8.5 kcal/mol; Drug120 also presents a similar range of binding affinities, though slightly lower. On the other hand, Drug73 displayed the best binding affinity (with mean docking scores above 10 kcal/mol) in comparison to the other two inhibitors.

To ascertain that the predicted poses interact with the activation loop and the leucine zipper, we also determined the type of interactions taking place between the studied ligands and ERK2 (Figure 6). Associations with H176 were predominant in all cases, mediated by van der Waals and hydrophobic interactions. Drug120 also presented an additional hydrogen bond with H176. H176 was not the only residue mediating in ERK2 binding; H178 and R170 were detected as potential interaction sites. In the case of H178, it produced van der Waals and hydrophobic contacts with DEL_22379 and Drug120, and also established hydrogen bonds, acting as a hydrogen-bond donor in both cases. Drug73, on the other hand, did not form hydrogen bonds. Moreover, Drug73 was the only one that interacted with F181. Residue F181, similarly to H176, is included in the dimerization interface as part of the activation loop. As such, it could also be relevant for dimerization inhibition.

In the three cases, at the dimerization interface, the activation loop was the most favored domain for ligand binding, as shown in Figure 7. However, the leucine zipper was also necessary for interaction with the candidate drugs. At the leucine zipper, L333 was the only residue that interacted with the three inhibitors. In general, the most common interaction type was van der Waals contact. Both Drug73 and Drug120 presented interactions with E332, van der Waals in both cases, and a hydrogen bond in the case of Drug120. Drug73 also interacts with adjacent residues than other candidates: DEL-22379 interacts with 5 residues and Drug120, with 7, while Drug73 interacts with 10 residues, such as D335 or F181 (Figure 6). These additional interactions could explain its higher binding affinity. Moreover, 2D structures of these inhibitors (Figure 4b) show that Drug73 differs more from DEL-22379 than Drug120, which also conserves the indoline-2-one group, which could be related to the similarity in binding energies that Drug120 and DEL-22379 presented. This could also explain the reason for the generation of other interactions that favor a higher affinity with the ERK2 dimerization interface.

### 2.5. Chemical Profile Predictions

Next, an in silico analysis was performed to obtain a preliminary prediction of the ADMET profile of the candidate compounds. To do so, the specific modules ProtoTOX and ProtoADME from the technological platform ProtoPRED (accessed 1 June 2025) were used for the prediction of a complete panel of relevant endpoints (including druglikeness, Caco-2 permeability, or half-life, among others), and Deep-PK for acute and chronic rat toxicity. We used these tools based on quantitative structure–activity relationship models (QSAR) (models implemented in ProtoPRED have been developed following the OECD principles) that present values above 0.6 in accuracy, sensitivity, and specificity in the validation set for classification models; and above 0.7 for regression models. The metric of the models can be found in the corresponding QSAR Model Reporting Format (QMRF) documents or available information in Appendix A. As can be seen in Table 1, all candidate compounds satisfied Lipinski’s rule of five, indicating no violations. Regarding oral bioavailability, the reference compound DEL-22379 was predicted as negative at both 20% and 30%, whereas Drug73 and Drug120 were positive in both thresholds. Caco-2 permeability values were lower for DEL-22379 (−5.34 log cm/s) compared to Drug73 (−4.89 log cm/s) and Drug120 (−4.78 log cm/s), in agreement with the predicted positive intestinal absorption for all three molecules. Predicted half-life values ranged between 4.4 and 7.1 h, with Drug73 showing the longest prevalence.

In terms of safety-related endpoints, DEL-22379 was classified as toxic in the acute oral toxicity model (300 mg/kg), while both Drug73 and Drug120 were predicted as non-toxic. Neurotoxicity estimations showed higher LD50 values for Drug73 (551.9 mg/kg) and Drug120 (497.1 mg/kg) compared with DEL-22379 (216.4 mg/kg). Similarly, rat acute toxicity (lethal dose 50; LD50) and chronic toxicity (Lowest observed adverse effect level, LOAEL) values were slightly more favorable for Drug73 and Drug120 relative to DEL-22379.

In addition, a BOILED-Egg model (Figure 8) was generated with the SwissAdme server (http://www.swissadme.ch/ (accessed 1 August 2025)) and utilized to visualize human intestinal absorption (HIA) and blood–brain barrier (BBB) penetration potential. All three compounds fall inside the white ellipse, indicating a positive prediction for human intestinal absorption in concordance with Table 1 results. Drug120 is located within the yellow area, suggesting a higher probability of brain penetration, while DEL-22379 and Drug73 are positioned outside this region. Regarding P-glycoprotein (P-gp) substrate status, DEL-22379 was predicted as P-gp positive, in contrast to Drug73 and Drug120, which were classified as non-substrates.

## 3. Discussion

The study of protein–protein interactions and their modulation has been the focus of numerous therapeutic-related studies [19,20]. In this work, we focused on the ERK2 dimerization process and its inhibition. This process is triggered by MEK-mediated phosphorylation at the TEY motif (T183-E184-Y185 in rats; T185-E186-Y186 in human ERK2) [21], after which ERK2 forms a homodimer. This homodimerization has an important role in various cellular processes, such as cell motility [11]. Likewise, it is relevant in some tumorigenesis processes [4,5,6].

As the MAPK/ERK pathway is central for the development of different tumor types, many compounds have been studied and subjected to clinical trials looking for means to inhibit aberrant activation of this cascade. As of today, here are some therapeutic compounds approved by the Food and Drug Administration (FDA) whose mechanism of action involves inhibiting components of the MAPK/ERK cascade, such as trametinib, a MEK inhibitor used for metastatic melanoma [22]. However, although ERK2 is a very relevant target in carcinogenesis, there is no approved inhibitor yet [23]. ERK2 inhibitor development has been quite challenging for researchers over the years. Several candidate compounds have been developed to bind to the catalytic site and act as ATP competitors, but these types of inhibitors have been hampered by toxicity and resistance mechanisms [24].

However, apart from the catalytic domain, ERK2 displays other domains amenable to being targeted for therapeutic intervention, such as D-recruiting and F-recruiting sites [25], or the already mentioned dimerization interface, which have been used as targets in different studies related to ERK2 inhibition. In our previous work [12], DEL-22379 was characterized as an ERK2 dimerization inhibitor, presenting the capacity to inhibit cell growth in RAS-ERK pathway-mutant tumor cells, without affecting ERK2 phosphorylation. Likewise, DEL-22379 displays activity in other ERK2 dimerization-related functions, such as in memory reconsolidation and synaptic plasticity, inhibiting hippocampal ERK2 dimerization in EGF-stimulated in vivo results [26]; or in anaplastic thyroid cancer induced by BRAF mutation [13].

DEL-22379 has been reported to bind to the ERK2 dimerization interface [12], formed by the activation loop, which includes residues H176 and F181; and a leucine zipper, formed by residues F329, E332, L333, D335, L336, P337, E339, K340, and E343 [27]. A previous work of an H176E and L_4_A ERK2 mutant unveiled the fact that the repulsive forces generated between residues H176E and E343, and the lack of hydrophobic forces that form the leucine zipper, were enough to disrupt the ERK2 dimer [27]. Therefore, targeting these residues could be essential for preventing ERK2 dimerization.

The aim of this study was the identification of DEL-22379-derived compounds to obtain optimized effects on the prevention of ERK2 dimerization. Yang and collaborators [16] carried out a study in 2020, in which they obtained DEL-22379 derivatives through experimental modifications of the chemical structure. They preserved the indolin-2-one scaffold, favoring hydrophobic and pi–pi stacking interactions among F181 and F329, and the benzene ring present in the scaffold. However, their effect on ERK dimerization was never tested directly [18]. In our case, we generated the derivatives through scaffold hopping, an in silico approach that allows the development of several modifications on a chemical scaffold in a short period of time [28]. Therefore, we could generate a large dataset with different variations from the original scaffold, including modifications on the indolin-2-one. As a result, we obtained 536 candidates, of which we only kept 300, based on their distance to H176 after a quick virtual screening. This way, we could avoid compounds that bind to regions far from the dimerization interface, thus avoiding false positives.

The use of in silico approaches, such as virtual screening, allows testing of a large number of molecules in just a few hours. This provides a reduced set of compounds to be experimentally validated [29]. They not only provide information about possible interactions with the therapeutic target, but also information about their mechanism of action [30]. After experimental evaluation, we could identify two novel potential inhibitors: Drug73 and Drug120. Both compounds were capable of inhibiting ERK2 dimerization without inhibiting its phosphorylation, which resembles DEL-22379’s effect. Knowing that DEL-22379 binds to the dimerization interface, we used in silico approaches to study the possible mechanism of action of these candidate inhibitors.

Molecular docking allowed us to predict the binding modes of these ligands in complex with the target, and therefore, to study the predicted interactions. To carry out the docking simulations between ERK2 and the hit compounds, we retrieved the phosphorylated crystalized structure of ERK2 (PDB ID: 2ERK). The activation loop, which is part of the dimerization interface, changes its conformation depending on ERK2 phosphorylation state, which can be unphosphorylated, monophosphorylated, or dual phosphorylated [31]. H176, as we mentioned previously, has been defined as one of the most relevant residues for ERK2 dimerization [27]. As DEL-22379 interacts with H176, we wanted to check whether the inhibitor candidates did as well. After analyzing the molecular docking results, we checked that both candidate compounds interacted with H176 residue in most of the poses, demonstrating the reproducibility of this interaction, and thus proving that these compounds could be optimal ERK2 dimerization inhibitors. In addition, Drug73 presents a higher binding energy than DEL-22379, suggesting that this new candidate could be an optimized structure of DEL-22379.

Although our results showed good values in binding affinity, it is not sufficient to consider these compounds as ERK2 dimerization inhibitors or as therapeutic compounds. A total of 90% of the compounds that enter the clinical phase fail to be developed as drugs [32]. To avoid drug discovery failure, it is important to consider the ADMET profile of the candidate compounds [33]. To obtain the ADMET profile, as well as the toxicity of these molecules, we used a computational tool based on quantitative structure–activity relationship models. This tool allowed us to analyze the chemical profile of the candidates, showing that both Drug73 and Drug120 presented better oral bioavailability in comparison with DEL-22379. These results also showed that DEL-22379 presented acute oral toxicity and a lower LD50 than the new candidate compounds in both neurotoxicity and acute toxicity in rat models.

This study focuses primarily on the molecular and biochemical characterization of ERK2 dimerization inhibitors and does not assess the downstream phenotypic effects resulting from ERK2 inhibition. Evaluating such functional consequences, for example, on cell migration, proliferation, or viability, would help to better guide the design and optimization of future ERK2 dimerization inhibitors. Future studies will address these aspects to confirm the biological and therapeutic significance of the candidate compounds. These results, hence, presented the identification of new hit compounds using a combined in silico and in vitro workflow. In this case, we presented that Drug73 and Drug120 could act as ERK2 dimerization inhibitors, with even better results as DEL-22379, showing that in silico pipelines can be used in drug discovery and drug optimization, and in combination with experimental evaluation, results can be obtained in a shorter period of time and at a lower cost.

## 4. Materials and Methods

### 4.1. Scaffold Hopping and Virtual Screening

For scaffold hopping, we used a web tool belonging to Mcule (https://mcule.com/, accessed on 1 May 2021) [34]. The scaffold hopping technique is used to obtain derivative chemical compounds of a primary structure by changing functional groups. This technique is used to generate a dataset of compounds that present similarities with DEL-22379. Once the dataset was generated, we performed a virtual screening with molecular docking to validate the affinity of the candidate compounds with the target protein ERK2. Virtual screening was carried out with YASARA software (v.21.6.17) [35,36,37]. Once the distance was obtained, the dataset was filtered. All compounds that were a distance higher than 5 Å were removed. We calculated the MACCS fingerprint to evaluate the Tanimoto similarity scores using the RDKit Python package (v.2022.03.5) [38].

### 4.2. Experimental Evaluation Protocol

For the experimental evaluation, we treated HEK293T cells with candidate compounds (10 µL) for 30 min, and then stimulated them with EGF to activate the MAPK/ERK cascade (100 ng/mL for 10 min). A native PAGE was carried out to identify the presence of dimers in cells, following a previously described protocol [39], while SDS-PAGE was used to evaluate ERK phosphorylation status. For the quantitative analysis of band intensity, we used Fiji software (v.17.02.0) [40], using the mean gray value.

### 4.3. Detailed Molecular Docking

For detailed molecular docking development, we first retrieved phosphorylated ERK2 protein from PDB (https://www.rcsb.org/, accessed on 10 November 2025) [41] (PDB ID: 2ERK; organism: *Rattus norvegicus*). ERK2 is conserved across different species, including *Homo sapiens*. We initially prepared the structure with PDB2PQR v3.7.1 [42], a biomolecular structure conversion software used to set the adequate protonation state of residues at biological conditions, setting a pH of 7.5, similar to the cytosolic pH. Water removal and hydrogen addition were performed with the AutoDock Tool (MGLTOOLS v.1.5.7) [43]. Ligand structures were generated with RDKit (v.2022.03.5) [38]. Once the structures were prepared, we used AutoDock Vina [44] available on YASARA (v.21.6.17) to predict the interaction modes of the hit molecules with the ERK2 protein. We directed the docking simulation to the dimerization interface residues (H176, F181, F329, E332, L333, D335, L336, P337, E339, K340, and E343), with a cubic simulation cell of 5Å around the dimerization residues (grid size of 38.97 Å in each axis) and setting the dimerization interface residues as flexible. Fifty poses were generated for each ligand, which were postprocessed through clustering, considering molecules from the same cluster when the RMSD value is not higher than 3 Å.

### 4.4. Protein-Ligand Interaction Analysis

All docking poses for each ligand were analyzed with in-house Python [45] scripts, using MDAnalysis (v.2.7.0) [46,47], ProLIF (v.2.0.3) [48], and RDKit (v.2022.03.5). MDAnalysis was used to load and prepare structures for interaction calculation, measured with ProLIF. Two-dimensional structures were generated with RDKit. Results visualization was carried out with ChimeraX (v.1.10.1) [49] and matplotlib (v.3.4.2) [50].

### 4.5. Chemical Profile Predictions Procedure

SMILES notation for DEL-22379, Drug73, and Drug120 was used to perform the in silico predictions of different toxicological and ADME properties by means of ProtoPRED server v.1.0 (https://protopred.protoqsar.com/, accessed on 10 August 2025) [51] and Deep-PK (https://biosig.lab.uq.edu.au/deeppk/, accessed on 10 August 2025) [52]. Moreover, a Boiled-Egg plot was performed using the SwissAdme web server (http://www.swissadme.ch/, accessed on 10 August 2025) [53].

## 5. Conclusions

In this study, we followed a workflow based on a combination of in silico and in vitro techniques to identify compounds capable of inhibiting ERK2 dimerization without affecting phosphorylation. We used computational techniques, such as scaffold hopping and virtual screening, to generate a dataset of derivative compounds that could act as ERK2 dimerization potential inhibitors from DEL-22379, a known dimerization inhibitor. These candidate compounds were screened on HEK293T to evaluate the inhibitory capacity of the dataset. Two hit compounds, Drug73 and Drug120, produced the same effects on the tested cells as control DEL-22379. Thus, we studied the potential mechanism of action of these compounds through more detailed molecular docking and ADMET prediction. Drug73 presented a higher affinity with ERK2 than DEL-22379, and both new candidates presented more drug-like properties based on a chemical profile study. This research demonstrated that the use of computational methods can accelerate and facilitate the search for new compounds, obtaining optimized inhibitors based on DEL-22379.

## Figures and Tables

**Figure 1 ijms-26-11481-f001:**
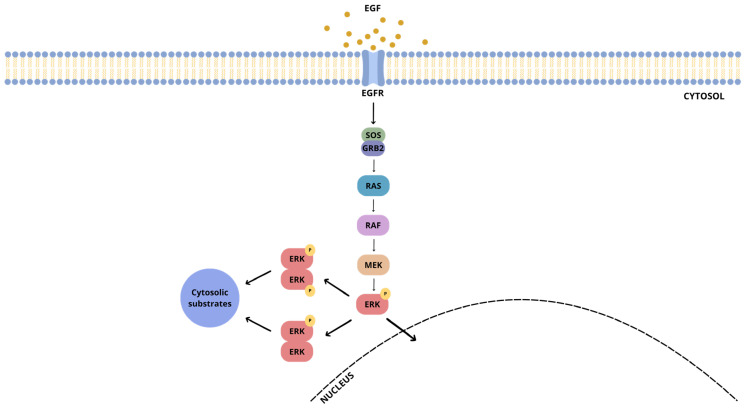
Schematic representation of the ERK1/2 cascade.

**Figure 2 ijms-26-11481-f002:**
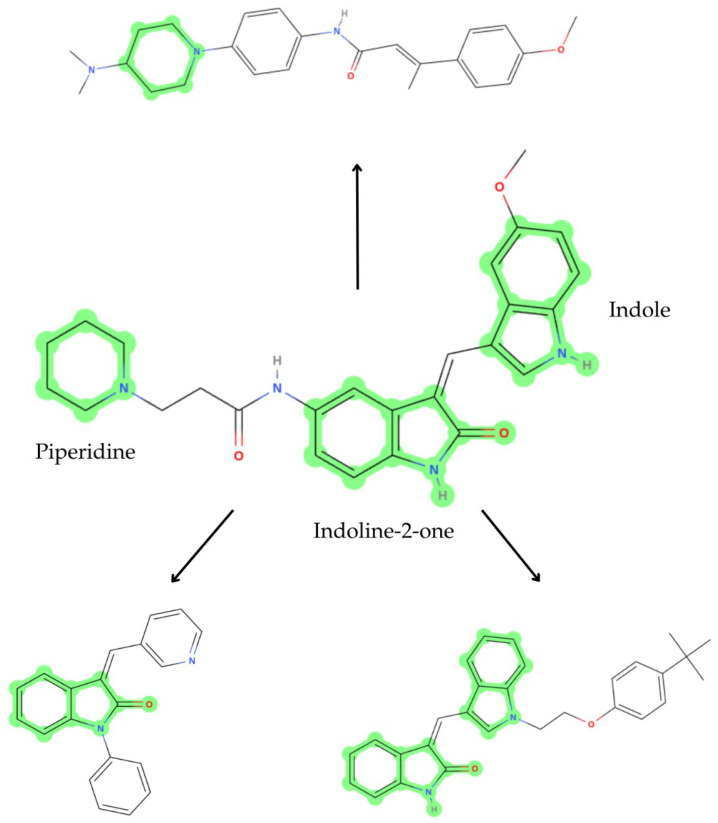
Two-dimensional representation of DEL-22379 and some derivative compounds showing the different scaffolds (highlighted in green). The DEL-22379 structure is represented in the middle. Selected scaffolds indicated on the DEL-22379 structure: indole on the right; indoline-2-one in the middle; piperidine on the left.

**Figure 3 ijms-26-11481-f003:**
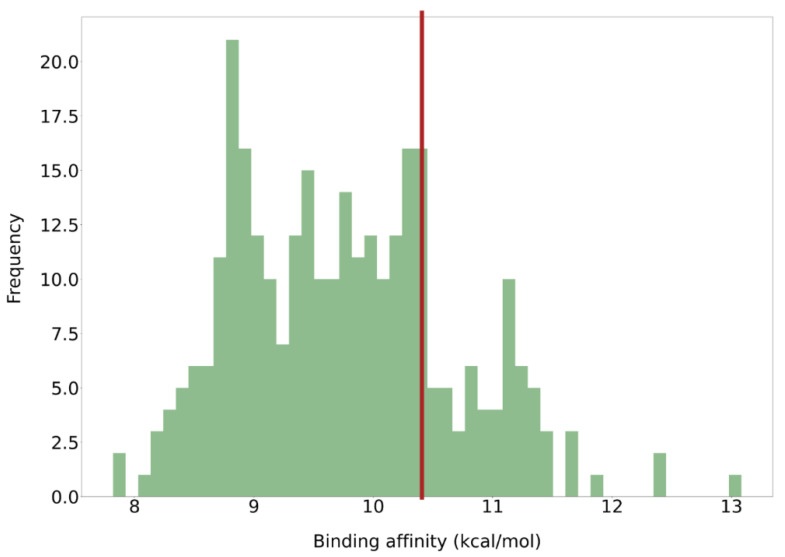
Binding affinity distribution of the dataset. According to the YASARA software, the higher the binding affinity, the better the result. The red line represents a threshold set by the docking score of DEL-22379 (10.461 kcal/mol; red line) as a reference.

**Figure 4 ijms-26-11481-f004:**
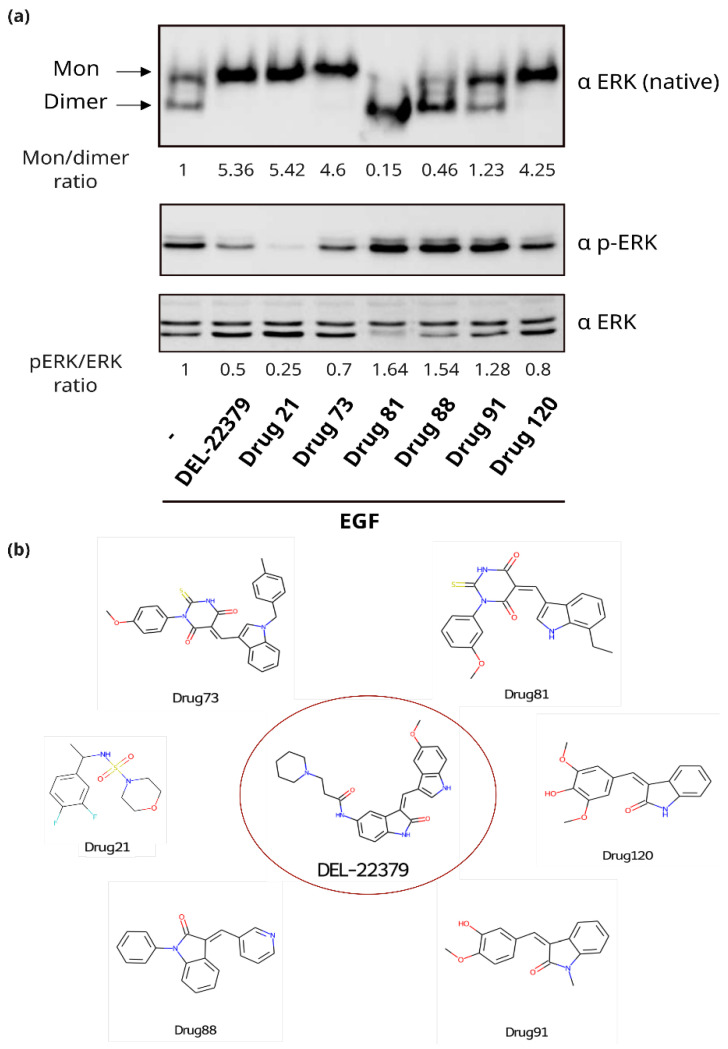
Identification of compounds that effectively inhibit ERK2 dimer formation without affecting ERK2 phosphorylation. (**a**). Native PAGE and SDS-PAGE of different tested compounds. HEK293 cells were stimulated with EGF (100 ng/mL for 10 min) and treated with different compounds or DEL-22379 as the positive control (10 μM for 30 min); protein levels were quantified from Western blots, and ratios are shown in the figure. (**b**). Two-dimensional structure of the shown compounds.

**Figure 5 ijms-26-11481-f005:**
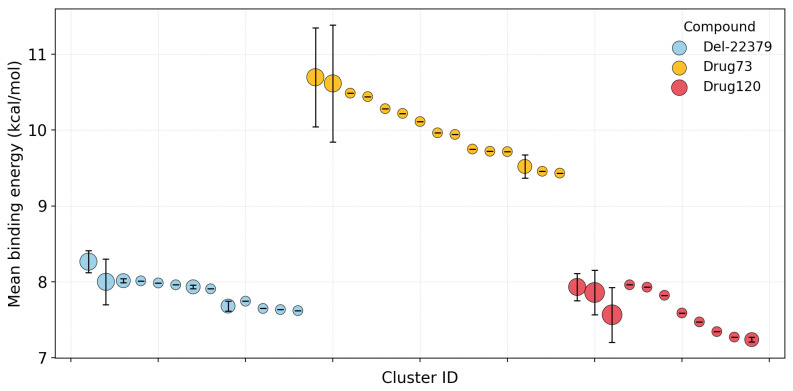
Mean binding affinity per cluster. The size of circles depends on cluster size.

**Figure 6 ijms-26-11481-f006:**
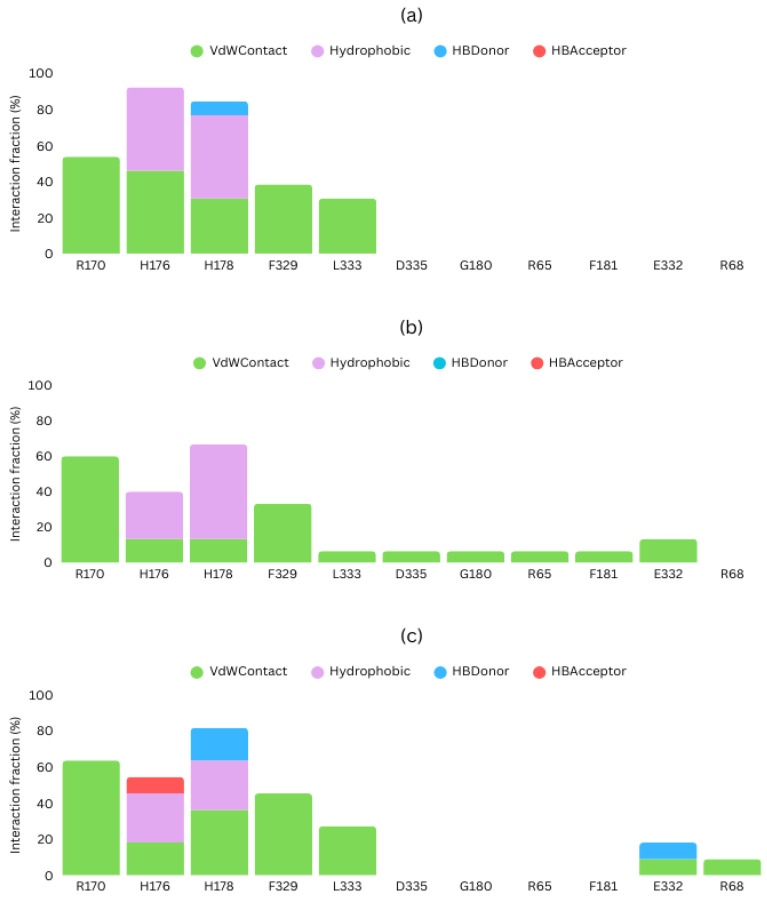
Types of interaction present between inhibitor candidates and ERK2. (**a**) DEL-22370; (**b**) Drug73; (**c**) Drug120.

**Figure 7 ijms-26-11481-f007:**
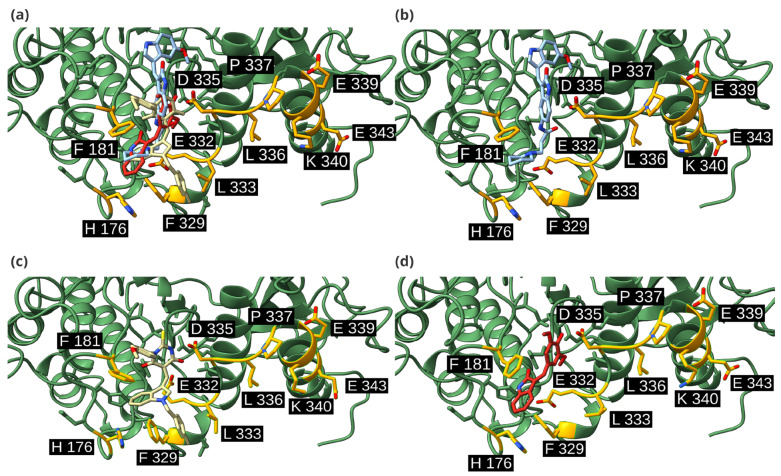
In silico representation of the best binding mode for each ligand in complex with ERK2. Dimerization interface represented in orange; DEL-22379 represented in cyan; Drug73 in yellow; Drug120 represented in red. (**a**) All inhibitors in complex with ERK2; (**b**). DEL-22379; (**c**). Drug73; (**d**). Drug120. Visualization on ChimeraX.

**Figure 8 ijms-26-11481-f008:**
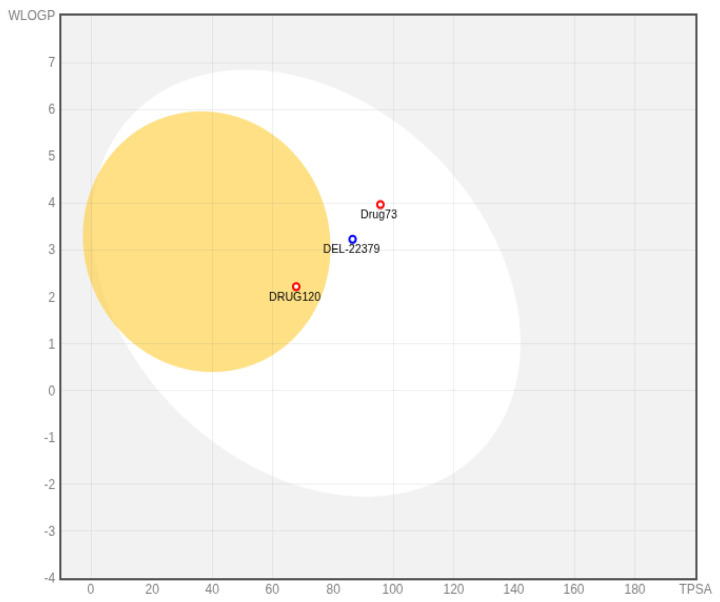
Boiled egg plot for the three studied compounds. Yellow: BBB+; white: HIA+; Blue dot indicates P-gp positive prediction while red dots indicates P-gp negative predictions.

**Table 1 ijms-26-11481-t001:** In silico predictions of ADMET properties employing ProtoPRED. Abbreviations: HIA, human intestinal absorption; LD50, lethal dose 50; LOAEL, lowest observed adverse effect level.

ADMET Properties	DEL-22379	Drug73	Drug120
Lipinski rule violations	0/4	0/4	0/4
Bioavailability 20%	Negative	Positive	Positive
Bioavailability 30%	Negative	Positive	Positive
Caco-2 permeability (log cm/s) 4	−5.34	−4.89	−4.78
HIA	Positive	Positive	Positive
Half-life (h)	6.60	7.1	4.4
Acute oral toxicity (300 mg/kg)	Toxic	Non-toxic	Non-toxic
Neurotoxicity [LD50] (mg/kg)	216.4	551.9	497
Acute toxicity in rats [LD50]	1.9	2.57	2.32
Chronic toxicity in rats (LOAEL)	2.69	1.83	2.07

## Data Availability

The original contributions presented in this study are included in the article/Appendix A. Further inquiries can be directed to the corresponding authors.

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
