# Peer review of "Identification and Characterization of ERK2 Dimerization Inhibitors by Integrated In Silico and In Vitro Screening"

_ijms, 2025, doi:10.3390/ijms262311481_

Round 1
Reviewer 1 Report
Comments and Suggestions for Authors
This study integrates scaffold hopping, virtual screening, and in vitro assays to identify new ERK2 dimerization inhibitors, highlighting Drug73 and Drug120 as promising analogs of DEL-22379 with better predicted pharmacokinetic and toxicity profiles. The work is original and relevant, presenting a coherent in silico-experimental workflow. However, the study would benefit from more rigorous docking validation, quantitative analysis of electrophoresis data, and additional biological assays to confirm functional relevance. Overall, it is a well-structured and valuable contribution that requires moderate methodological strengthening before publication.
Major suggestions:
- The manuscript would benefit from a clearer explanation of the scaffold hopping strategy, including chemical rationale, diversity metrics, and how analogs were prioritized beyond H176 distance filtering.
- The molecular docking protocol requires more methodological rigor details such as grid dimensions, scoring function validation, and reproducibility assessment (e.g., redocking of DEL-22379 or RMSD comparison) should be included.
- The experimental validation (native PAGE) lacks quantitative analysis.
- The functional significance of inhibition should be further validated, for example by testing cell migration or viability assays, to connect biochemical inhibition with phenotypic outcomes.
- The ADMET predictions, while informative, should be cautiously interpreted and, if possible, supported by at least one experimental cytotoxicity or solubility assay.
- Figures 5-7 could be enhanced with clearer residue labels and consistent color coding of ligands and binding sites to improve interpretability.
Minor suggestions:
- Several typographical and formatting inconsistencies should be corrected.
- Abbreviations such as HIA, LOAEL, and P-gp should be defined on first use in the main text.
- The Results section could be shortened by merging overlapping descriptions of docking and ADMET predictions.
Reviewer 2 Report
Comments and Suggestions for Authors
Crespo and co-workers reported a combination of computational and experimental techniques to identify suitable compounds that could inhibit ERK2 dimerization without affecting phosphorylation. The manuscript is well written, and the results are presented clearly. The authors used oxindole-based compounds as references and designed new derivatives containing heterocyclic moieties to control bioactivity. Among the various drug candidates investigated in this study, two showed particularly promising results. Such studies are important for drug development, and the growing role of computational methods warrants the publication of this work in the IJMS journal.
However, the introduction of the various drug names in Figure 4 is somewhat unclear. It is advisable to include the structures of these drugs at the point in the manuscript where they are first introduced. For example, in Figure 7, the authors have provided the structures of the most effective drug candidates—similar trend should be given to the other drug structures as well.
In Figure 5, drug 73 showed better binding affinity than drug 120. Could the authors rationalize the reason for this difference?
